# Impact of Smoking Status on Lung Cancer Characteristics and Mortality Rates between Screened and Non-Screened Lung Cancer Cohorts: Real-World Knowledge Translation and Education

**DOI:** 10.3390/jpm12010026

**Published:** 2022-01-02

**Authors:** Fu-Zong Wu, Yun-Ju Wu, Chi-Shen Chen, Shu-Ching Yang

**Affiliations:** 1Institute of Education, National Sun Yat-sen University, 70, Lien-Hai Road, Kaohsiung 80424, Taiwan; cmvwu1029@gmail.com; 2Department of Radiology, Kaohsiung Veterans General Hospital, Kaohsiung 81362, Taiwan; yjwu@vghks.gov.tw; 3Department of Medical Research and Education, Kaohsiung Veterans General Hospital, Kaohsiung 81362, Taiwan; 4Faculty of Medicine, School of Medicine, National Yang Ming Chiao Tung University, Taipei 11221, Taiwan; 5Faculty of Clinical Medicine, National Yang Ming Chiao Tung University, Taipei 11221, Taiwan; 6Physical Examination Center, Kaohsiung Veterans General Hospital, Kaohsiung 81362, Taiwan; chirise@vghks.gov.tw

**Keywords:** lung cancer screening, smoking, willingness to screen, education

## Abstract

This was a retrospective hospital-based cohort study of participants diagnosed with lung cancer in the lung cancer register database, and our goal was to evaluate the impact of smoking and screening status on lung cancer characteristics and clinical outcomes. According to the hospital-based lung cancer register database, a total of 2883 lung cancers were diagnosed in 2883 patients between January 2007 and September 2017, which were divided into four groups according to smoking and screening status. A comparison was performed in terms of clinical characteristics and outcomes of lung cancer between the four groups. For non-smokers, age, gender, screened status, tumor size, targeted therapy, and curative surgery were independent prognostic factors of overall survival for lung cancer subjects. However, screened status and gender were not significant prognostic factors for lung cancer survival in smokers with lung cancer. For the non-smoker group, about 4.9% of lung cancer subjects (N = 81) were detected by screening. However, only 0.97% of lung cancer subjects (N = 12) were detected by screening in smokers. This could be attributed to smokers’ negative attitudes and low socioeconomic status preventing LDCT lung cancer screening. In summary, our real-world data suggest that effectively encouraging smokers to be more willing to participate in lung cancer screening programs with screening allowance and educational training in the future is an important issue.

## 1. Introduction

Lung cancer is the most common cause of cancer-related death among both men and women worldwide in recent years, and about 70% of lung cancers are diagnosed at an advanced stage with poor survival outcomes [1,2,3]. Recent systematic reviews and meta-analyses across 31 randomized control trials (RCTs) have demonstrated the prolonged, substantial survival benefit of low-dose computed tomography (LDCT) for lung cancer screening in high-risk heavy smokers, with a 20% reduction in lung-cancer-specific mortality compared with the control group [4].

In recent years, several previous studies have emphasized the importance of a lung cancer screening program in the population with a high prevalence of non-smoking-related lung cancer (reported prevalence of 1.55 to 2.34%) [5,6,7,8,9,10]. These studies have addressed that implementing a mass LDCT lung cancer screening program targeting the high-risk population could reduce the lung cancer mortality rate in the population with a high prevalence of non-smoking-related lung cancer.

Smoking causes genetic damage to epithelial cells in the lung and suppresses immune surveillance, which leads to an increased risk of lung cancer development [11,12]. Therefore, smoking is a well-known major risk for lung cancer development.

To the best of our knowledge, few studies have investigated the interaction of smoking and screening status on lung cancer characteristics and mortality rate. Therefore, this study was designed to provide an integrated approach to studying the relationship among smoking status, screening status, willingness to screening, and lung cancer characteristics with the prognostic outcome of subjects in a hospital-based lung cancer register cohort.

## 2. Materials and Methods

### 2.1. Data Collection

This was a retrospective cohort study based on the hospital-based lung cancer register database at Kaohsiung Veterans General Hospital between January 2007 and September 2017. The focus in this present study is on clinical characteristics and outcomes of lung cancer between screened and non-screening patient groups. The Institutional Review Board of Kaohsiung Veterans General Hospital approved this retrospective study, and thus informed consent was waived (VGHKS19-CT2-09). All research was performed in accordance with the relevant guidelines and regulations and all Institutional Review Board requirements. The study included 2883 patients aged 40–80 years with lung cancer diagnosed at Kaohsiung Veterans General Hospital, divided into four groups according to smoking and screening status. The study flowchart is illustrated in Figure 1. Patients were classified into four groups: group 1 included 1570 lung cancer subjects who had never been screened for lung cancer with LDCT and had no history of smoking; group 2 included 81 lung cancer subjects who had been screened for lung cancer with LDCT and had no history of smoking; group 3 included 1220 lung cancer subjects who had never been screened for lung cancer with LDCT and had a history of smoking, and group 4 included 12 lung cancer subjects who had been screened for lung cancer with LDCT and had a history of smoking. The subjects were classified according to the stringent definition of screening criteria. Positive screening status was defined as patients who had no clinical symptoms but had incidentally detected lung cancers by a self-paid LDCT exam between the ages of 40–80 years. Non-smoker was defined as an adult who had never smoked or who had smoked less than 100 cigarettes in his or her lifetime. Positive smoking history was defined as current smokers or a cessation of smoking within the previous 5 years, and who had smoked at least 100 cigarettes in his or her lifetime according to National Health Interview Survey. The information on survival and lung cancer deaths is based on the hospital-based center register data.

Clinical characteristics were recorded and compared for analysis in the four groups. The hospital-based lung cancer register database included the following patient demographics and clinical characteristics that are summarized in Table 1 (age, sex, tumor size, histopathologic type, adenocarcinoma spectrum classification, lung cancer clinical stage, lung cancer death, survival time, mortality rate, curative surgical treatment, targeted therapy, smoking habit, betel nut consumption, and alcohol drinking habit). The histological diagnosis was described according to the World Health Organization classification. All the patients were staged according to the seventh edition of the TNM staging system [13].

### 2.2. Statistical Analysis

The patient demographics and clinical characteristics are expressed as mean ± standard deviation (SD) or median (interquartile range, IQR) and frequency (%) for group comparison. Multiple group comparisons were completed by analysis of variance (ANOVA) for normally distributed data and Kruskal–Wallis test for skewed data. We used the Kolmogorov test to assess the normality assumption in analysis of variance. We used the post hoc Bonferroni test to analyze the differences between each group. Cox regression analysis was performed to calculate and compare the survival rate between screened and non-screening groups in smokers and non-smokers. The proportional hazards assumption refers to the fact that the hazard functions are multiplicatively related. A multivariate analysis was used to estimate the hazard ratios (HRs) based on the Cox regression model, adjusted for age, gender, smoking, alcohol consumption, betel nut consumption, tumor size, curative surgery, targeted therapy, histology, and screening status. All statistical analyses were performed with SPSS 18.0 for Windows (SPSS Inc, Chicago, IL). A *p*-value of 0.05 was regarded as significant.

## 3. Results

The baseline characteristics of lung cancer subjects classified into four groups according to the screened and smoking status are shown in Table 1. There were statistically significant differences across the four groups for the clinical characteristics and outcomes demonstrated by one-way ANOVA. Bonferroni post hoc test results (inter-group comparison) are also shown in Table 1.

### 3.1. Non-Smokers in Two-Group Comparison

For non-smokers, the general characteristics and outcomes of the lung cancer subjects between the screened and non-screened groups are summarized in Appendix A. Table 2 summarizes the mortality and survival analysis according to screened and non-screened status for non-smokers. For non-smokers, patients in the screened group had lower overall mortality than those in the non-screened group. One- and five-year mortality rates increased significantly from the screened group to the non-screened group (*p* < 0.001 for all). The mean survival time for the screened group was 892.05 ± 516.24 days (median 825 days), and the mean survival for the non-screened group was 676.03 ± 600.47 days (median 517.5 days).

Cox regression was used to examine the association between screening status and survival and identify predictors of survival among non-smokers. For non-smokers, in the survival analysis for lung cancer subjects using the Cox regression model for multivariate effects, the hazard ratio for lung cancer mortality was determined adjusting for age, gender, smoking, alcohol consumption, betel nut consumption, tumor size, curative surgery, targeted therapy, histology, and screening status (Table 3). Multivariate survival analysis using Cox’s regression model showed that age (HR = 1.011, *p* < 0.001) and tumor size (HR = 1.012, *p* < 0.001) were associated with unfavorable survival for lung cancer subjects, as shown in Table 3. However, Cox’s regression model showed that a screening status (HR = 0.480, *p* = 0.040), gender (HR = 0.861, *p* = 0.034), targeted therapy (HR = 0.839, *p* = 0.031), and curative surgery (HR = 0.196, *p* < 0.001) were identified as independent, favorable prognostic factors of overall survival for lung cancer subjects, as shown in Table 3.

### 3.2. Smokers in Two-Group Comparisons

For smokers, the general characteristics and outcomes of the lung cancer subjects between the screened and non-screened groups are summarized in Appendix A. Table 4 summarizes the mortality and survival analysis according to screened and non-screened status of smokers. For smokers, patients in the screened group had lower overall mortality than those in the non-screened group. One- and five-year mortality rates increased significantly from the screened group to the non-screened group (*p* < 0.001 for all). The mean survival time for the screened group was 646.08 ± 337.21 days (median 683 days), and the mean survival for the non-screened group was 444.85 ± 468.43 (median 304 days, *p* = 0.064). However, the *p*-value from the test may not reach statistical significance with a small sample size in the screened group (N = 12).

Cox regression was used to examine the association between screening status and survival and identify predictors of survival among smokers. For smokers, in the survival analysis for lung cancer subjects using the Cox regression model for multivariate effects, the hazard ratio for lung cancer mortality was determined adjusting for age, gender, smoking, alcohol consumption, betel nut consumption, tumor size, curative surgery, targeted therapy, histology, and screening status (Table 5). Multivariate survival analysis using Cox’s regression model showed that age (HR = 1.014, *p* < 0.001) and tumor size (HR = 1.011, *p* < 0.001) were identified as independent, unfavorable prognostic factors of overall survival for lung cancer subjects, as shown in Table 5. Multivariate survival analysis using Cox’s regression model showed that targeted therapy (HR = 0.792, *p* = 0.023) and curative surgery (HR = 0.202, *p* < 0.001) were identified as independent, favorable prognostic factors of overall survival for lung cancer subjects, as shown in Table 5.

## 4. Discussion

The focus of the current study was to investigate the relationship between smoking status, screening status, willingness to screen, and lung cancer characteristics and the prognostic outcome of subjects in the hospital-based lung cancer register cohort. In this study, we demonstrated five major findings. The first one is that 57% of cancer subjects are non-smoking-related lung cancers, according to the hospital-based lung cancer register cohort. The second finding is that about 4.9% of lung cancer subjects (N = 81) are detected by screening in non-smokers. Third, only 0.97% of lung cancer subjects (N = 12) are detected by screening in smokers. Fourth, age, gender, screening status, tumor size, targeted therapy, and curative surgery are independently significant prognostic factors for lung cancer survival in non-smokers. Fifth, age, tumor size, targeted therapy, and curative surgery are independently significant prognostic factors for lung cancer survival in smokers.

In this present study, more than half (57%) were non-smoking-related lung cancers, according to the hospital-based lung cancer register cohort. Our study result is in line with previous studies in Asian countries. This finding suggests an increasing trend in the prevalence of non-smoking-related lung cancer in the Asian population [14,15,16]. In addition, if we only adopted the National Lung Screening Trial (NLST) criteria in the Asian population, approximately 70% of lung cancer subjects might have been missed, according to a previous study in Japan [17]. Therefore, not only heavy smokers, but also non-smokers at high risk should be enrolled in the target population for the LDCT lung cancer screening program. For the non-smoker group, about 4.9% of lung cancer subjects (N = 81) are detected by screening in non-smokers. However, only 0.97% of lung cancer subjects (N = 12) are detected by screening in smokers. This could be contributed to smokers’ negative attitudes and low socioeconomic status preventing LDCT lung cancer screening [18,19,20], as shown in Figure 2. Previous studies have shown that heavy smokers are less willing to be screened for lung cancer than former or non-smokers, because smokers do not believe in the survival benefits of early stage diagnosis [18,19]. In addition, results from our previous studies show that only about 28.75% of the study cohort had a smoking habit in our self-paid LDCT screening program [9]. Only 15% of the self-paid LDCT screening groups were eligible for NLST criteria [9,21]. Among all-screen detected lung cancer subjects, up to 87.1% (N = 81) of lung cancer subjects were non-smokers in the current study. As the number of screen-detected lung cancer subjects in smokers is still small (N = 12), larger studies will be needed to further explore the cause in the future. The percentage of screened lung cancer subjects in the non-smoker group was significantly higher than that of those in the smoker group (4.9% versus 0.97%; *p* < 0.001, shown in Figure 2).

These findings suggested that making heavy smokers more willing to participate in lung cancer screening programs is an important issue in the future [22]. A previous study demonstrated educational disparities in smokers, regardless of age and gender [23]. In line with the previous studies, our study suggests that we should pay more attention to shared-decision-making educational plans in heavy smokers.

In this study, we aimed to investigate the difference of prognostic factors between smoker and non-smoker lung cancer subjects. In this study, a screened status is an important favorable prognostic factor for lung cancer in non-smokers. However, our study results did not find screened status to be an important prognostic factor for lung cancer in smokers. Recent studies have demonstrated that low-dose computed tomography lung cancer screening trials have revealed a reduction in lung cancer mortality in subjects with a smoking history [24,25]. In our study, only 0.97% of lung cancer subjects (N = 12) were detected by screening in smokers. The rate is much lower than that of non-smoking lung cancer patients participating in lung cancer screening, as shown in Figure 2. These findings suggest that increasing smokers’ participation in lung cancer screening will improve lung cancer survival. However, several studies have shown that smokers are less likely to participate in lung cancer screening programs due to their low socioeconomic status or low willingness to screen [18,19,20]. As a result, the self-paid lung cancer screening program in this study has difficulty achieving significant effects in smokers. Through policy subsidies and shared-decision-making educational practices, barriers can be overcome, and the rate of smokers receiving lung cancer screening can be effectively increased. Therefore, screening can gradually reduce the lung cancer mortality of smokers in the real world.

In the non-smoking group, we found that female gender was a favorable prognostic factor predicting better survival. This could be due to the different biologic mechanisms of lung cancer that depend upon smoking status [26,27]. Previous studies have demonstrated a high prevalence of non-smoking-related lung cancer in the Asian population in the screening program, especially in women, usually manifesting with persistent subsolid nodules with indolent behavior [9,28,29]. In addition, about 30% of screen-detected lung cancer subjects met the diagnostic criteria for multiple primary lung cancers, according to previous studies [30,31]. However, these non-smoking lung cancer subjects have a better prognostic outcome of lung cancer. On the other hand, screening may also cause overdiagnosis and overtreatment during the screening process [32,33]. To maximize the benefit of LDCT screening and minimize the potential harm in over-diagnosis and over-management among non-smokers, a lung-cancer-risk-prediction algorithm with shared decision-making plans for indeterminate pulmonary nodule management is mandatory for a successful LDCT lung cancer screening program [34,35]. For smokers, effectively encouraging smokers to be more willing to participate in lung cancer screening programs with screening allowances and educational training programs (quitting smoking) in the future is an important issue. Promoting a personalized lung cancer screening program requires the support of government policies, primary care physicians, and public education for the incorporation of patient lung cancer risk, preferences, and socioeconomic status to resolve the complex trade-offs in the screening process.

This study benefits from real-world data through the prolonged implementation of the self-paid LDCT lung cancer screening program in this hospital-based cohort. This study describes why and how more non-smokers (especially women) could benefit from being diagnosed with lung cancer at an early stage. However, there are three major limitations in this study that could be addressed in future research. First, this study did not explore differences in attitudes and behavior towards screening for lung cancer between smokers and non-smokers. Real-world data from this study show that the smoking group had a very low proportion (0.97%) of lung cancer detected by screening, and only 12 people were diagnosed in this group (group 4). Our data also indirectly reflect smokers’ negative attitudes and behavior towards participating in lung cancer screening. This finding is supported by previous studies that described that heavy smokers are less willing to be screened for lung cancer than former smokers or smokers [18,19]. Second, our self-paid, screened population may not represent the screening population eligible for NLST criteria. Therefore, the study result may not be generalized to the population at other institutions or with other demographics. Third, potential residual confounding in this study lowers the certainty of the evidence.

## 5. Conclusions

In summary, this study aimed at investigating the different impacts of smoking and screening status on lung cancer characteristics and mortality rates. For non-smokers, female gender and screened status are two independent, important, and favorable prognostic factors for lung cancer survival. However, these findings could not be demonstrated in smokers, which may be due to limited numbers of lung cancer subjects classified as smokers who have a negative attitude and low socioeconomic status preventing LDCT lung cancer screening. Effectively encouraging smokers to be more willing to participate in government-subsidized lung cancer screening programs in the future is an important issue.

## Figures and Tables

**Figure 1 jpm-12-00026-f001:**
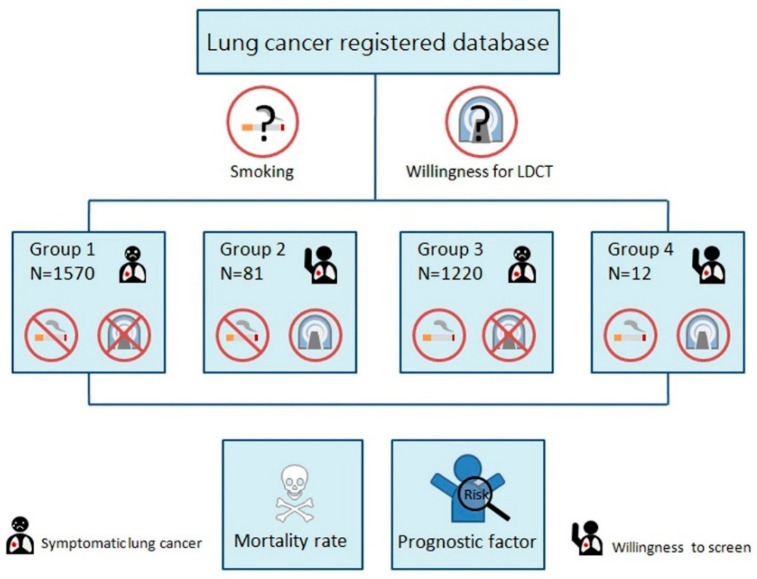
Flowchart of patient recruitment. Patients were classified into four groups according to smoking and screening status.

**Figure 2 jpm-12-00026-f002:**
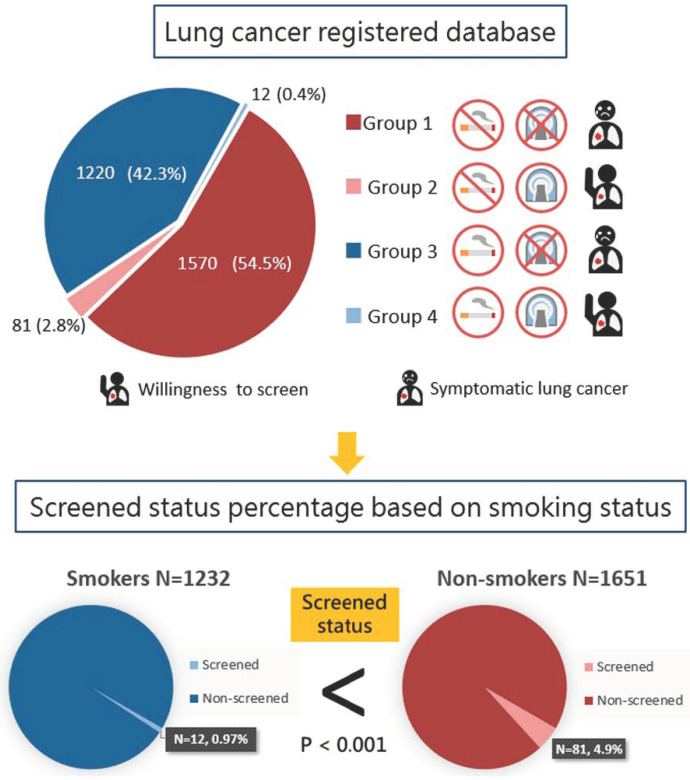
Screening status percentages according to lung cancer subjects based on smoking status.

**Table 1 jpm-12-00026-t001:** Demographic features of enrolled subjects for group (N = 2883) comparison based on smoking and screening status (means, standard deviations, and ANOVA results).

	Group 1 (N = 1570)	Group 2 (N = 81)	Group 3 (N = 1220)	Group 4 (N = 12)	*p*-Value	1 vs. 2	1 vs. 3	1 vs. 4	2 vs. 3	2 vs. 4	3 vs. 4
Mean age at diagnosis, years (mean, SD)	66.36 ± 12.78	59.41 ± 7.41	69.15 ± 13.00	63.33 ± 11.83	<0.0001	<0.0001	<0.0001	1	<0.0001	1	0.695
Median age at diagnosis, years (range)	66 (40–99)	66 (42–77)	71 (41–99)	72 (42–83)							
Gender (n, %)					<0.0001	0.437	<0.0001	<0.0001	<0.0001	<0.0001	1
Male	589 (37.5%)	24 (29.6%)	1175 (96.3%)	12 (100%)							
Female	981 (62.5%)	57 (70.4%)	45 (3.7%)	0 (0%)							
Smoking	0 (0%)	0 (0%)	1220 (100%)	12 (100%)							
Alcohol consumption	39 (2.5%)	4 (4.9%)	405 (33.2%)	6 (50%)	<0.0001	1	<0.0001	<0.0001	<0.0001	<0.0001	0.481
Betel nut consumption	8 (0.5%)	0 (0%)	187 (15.3%)	3 (25%)	<0.0001	1	<0.0001	0.003	<0.0001	0.005	1
Histology					<0.0001	0.009	<0.0001	1	<0.0001	0.613	1
Adenocarcinoma	1269 (80.8%)	79 (97.5%)	744 (61%)	9 (75%)							
Squamous cell carcinoma	170 (10.8%)	1 (1.2%)	273 (22.4%)	2 (16.7%)							
Small cell carcinoma	79 (5%)	1 (1.2%)	178 (14.6%)	0 (0%)							
Other	52 (3.3%)	0 (0%)	25 (2%)	1 (8.3%)							
Adenocarcinoma spectrum					<0.0001	<0.0001	1	1	<0.0001	<0.0001	1
AAH	0 (0%)	6 (7.4%)	0 (0%)	0 (0%)							
AIS	0 (0%)	7 (8.6%)	0 (0%)	0 (0%)							
MIA	0 (0%)	9 (11.1%)	0 (0%)	0 (0%)							
IPA	1269 (100%)	57 (70.4%)	744 (60.9%)	9 (75%)							
Stage					<0.0001	<0.0001	<0.0001	<0.0001	<0.0001	0.95	<0.0001
Carcinoma in situ	4 (0.3%)	14 (17.3%)	0 (0%)	0 (0%)							
I	326 (20.8%)	54 (66.7%)	129 (10.6%)	8 (66.7%)							
II	64 (4.1%)	4 (4.9%)	60 (4.9%)	1 (8.3%)							
III	334 (21.3%)	2 (2.5%)	302 (24.8%)	2 (16.7%)							
IV	842 (53.6%)	7 (8.6%)	729 (59.8%)	1 (8.3%)							
Curative surgery rate	459 (29.2%)	35 (83.3%)	211 (17.3%)	10 (83.3%)	<0.0001	<0.0001	<0.0001	<0.0001	<0.0001	1	<0.0001
Targeted therapy	319 (20.3%)	5 (11.9%)	197 (16.1%)	0 (0%)	0.008	0.982	0.028	0.418	1	1	0.898
Mean tumor size (mm)	41.25 ± 23.36	16.16 ± 13.74	51.12 ± 26.60	26.75 ± 19.99	<0.0001	<0.0001	<0.0001	0.249	<0.0001	0.976	0.004
Deaths	1159 (73.8%)	8 (9.9%)	1047 (85.8%)	2 (16.7%)	<0.0001	<0.0001	<0.0001	<0.0001	<0.0001	1	<0.0001
Mean survival days	676.03 ± 600.47	892.05 ± 516.24	444.85 ± 468.43	646.08 ± 337.21	<0.0001	0.003	<0.0001	1	<0.0001	0.869	1
Median survival days	517.5 (1–3128)	825 (30–2599)	304 (1–2937)	683 (22–1217)							

Abbreviations: AAH: atypical adenomatous hyperplasia; AIS: adenocarcinoma in situ; ANOVA: analysis of variance; IPA: invasive pulmonary adenocarcinoma; MIA: minimally invasive adenocarcinoma; SD: standard deviation.

**Table 2 jpm-12-00026-t002:** Clinical outcomes and mortality profiles of non-smokers with lung cancer according to screening status.

	Screened Group	Non-Screened Group	*p*-Value
Patients	N = 81	N = 1570	
Deaths	N = 8	N = 1159	<0.001
1-year mortality	1.25%	33.78%	<0.001
5-year mortality	15.55%	74.22%	<0.001
Overall mortality	9.90%	73.80%	<0.001
Average survival days	892.05 ± 516.24	676.03 ± 600.47	<0.001

**Table 3 jpm-12-00026-t003:** Multivariate Cox regression model of prognostic factors for non-smokers with lung cancer.

Variable	Hazard Ratio	95% CI	*p*-Value
Age	1.011	1.005–1.016	<0.001
Gender	0.861	0.750–0.989	0.034
Alcohol consumption	1.029	0.575–1.841	0.924
Betel nut consumption	0.879	0.315–2.453	0.805
Screened	0.480	0.238–0.967	0.040
Tumor size	1.012	1.009–1.015	<0.001
Targeted therapy	0.839	0.716–0.985	0.031
Histology	0.872	0.740–1.029	0.105
Curative surgery	0.196	0.162–0.238	<0.001

Abbreviations: CT: confidence interval; histology: adenocarcinoma versus other histology types (reference); gender (reference group: male); alcohol consumption (reference group: no alcoholic drinks); betel nut consumption (reference group: no betel nut consumption); screened (reference group: unscreened status); targeted therapy: (reference group: no targeted therapy); curative surgery (reference group: no curative surgery).

**Table 4 jpm-12-00026-t004:** Clinical outcomes and mortality profiles of smokers with lung cancer according to screening status.

	Screened Group	Nonscreened Group	*p*-Value
Patients	N = 12	N = 1220	
Deaths	N = 2	N = 1047	<0.0001
1-year mortality	8.33%	51.51%	<0.001
5-year mortality	17.50%	85.5%	<0.001
Overall mortality	16.7%	85.8%	<0.001
Average days of survival	646.08 ± 337.21	444.85 ± 468.43	0.064

**Table 5 jpm-12-00026-t005:** Multivariate Cox regression model of prognostic factors of smokers with lung cancer.

Variable	Hazard Ratio	95% CI	*p*-Value
Age	1.014	1.009–1.020	<0.001
Gender	0.788	0.536–1.158	0.225
Alcohol consumption	1.095	0.931–1.287	0.272
Betel nut consumption	0.912	0.733–1.135	0.409
Screened	0.386	0.096–1.553	0.180
Tumor size	1.011	1.008–1.013	<0.001
Targeted therapy	0.792	0.648–0.968	0.023
Histology	1.021	0.878–1.187	0.788
Curative surgery	0202	0.159–0.256	<0.001

Abbreviations: CT: confidence interval; histology: adenocarcinoma versus other histology types (reference); gender (reference group: male); alcohol consumption (reference group: no alcoholic drinks); betel nut consumption (reference group: no betel nut consumption); screened (reference group: unscreened status); targeted therapy: (reference group: no targeted therapy); curative surgery (reference group: no curative surgery).

## Data Availability

The data presented in this study are available on request from the corresponding author.

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
