# Peer review of "Impact of Smoking Status on Lung Cancer Characteristics and Mortality Rates between Screened and Non-Screened Lung Cancer Cohorts: Real-World Knowledge Translation and Education"

_jpm, 2022, doi:10.3390/jpm12010026_

Round 1

Reviewer 1 Report

The authors present here a retrospective study to investigate the impact of population characteristics and LDCT screening on the prognosis of lung cancer, depending on whether patients smoke or not.

LDCT  lung cancer screening studies are of major importance as lung cancer is a leading cause of cancer mortality worldwide.

However, there are substantive problems that make this work, although interesting, difficult to publish in its current state. 

Major remarks:

The title is problematic as it stands. It is particularly long, and does not specify the interest of the study presented. It should be shorter and more concise.

The rationale of the study is problematic. Studying the predictive characteristics of lung cancer prognosis is certainly of major interest, but here it is a question of studying the impact of, for example, tumour size on prognosis. However, the impact of such data is already known in the literature, since TNM classifications of tumors are directly linked to patient survival data. Moreover, the 7th classification was used here, whereas the most recent is the 8th; the retrospective nature of the study does not pose any problem in using an up-to-date classification.

 The statistical analyses are problematic. Indeed, statistical data were obtained from the comparison of data from cohorts of extremely heterogeneous power, with more than 1000 patients in the "no LDCT screening" cohorts, and less than 20 patients in the "LDCT screening" cohorts.
The definition of "LDCT screening" is also problematic, as it involves comparing data between patients screened by their own choice, whether or not they are smokers. The screening programs under study do not depend on individual choice, and target mainly smoking patients, which poses a major problem regarding the rationale of the study.
Maybe the rationale of the study should be reworked.

Some formulation are confusing regarding the litterature known datas. For example: "Smoking could cause genetic damage of epithelial cell": could is not appropriated here. This data is well demonstrated in the litterature and seems to be questioned here.

The conclusions currently presented are confusing, and do not allow a clear message to be drawn from this work, which is nevertheless consequent.

Minor remarks:

Many syntax errors are present.
exemple: "participants who diagnosed with lung cancer in the lung cancer register database" ligne 17
"effectively" instead of "indeed", line 30

"Patients were classified four groups": there is no subject in this sentence

Author Response

TITLE: RE:

RE: jpm-1448189, entitled "Impact of smoking status on lung cancer characteristic and mortality rate between screened and non-screened lung cancer cohorts: willingness and educational concern in lung cancer screening program"

We are grateful to the Reviewers for their valuable comments and suggestions. We have now addressed all of the Reviewer’s comments, and provide a point-by-point response below. All changes are underlined in the text of the revised manuscript.

Reviewer#1

Comment #1

LDCT lung cancer screening studies are of major importance as lung cancer is a leading cause of cancer mortality worldwide. However, there are substantive problems that make this work, although interesting, difficult to publish in its current state.

Author’s response:

We highly appreciate the reviewer’s comment. We are grateful to the Reviewers for their valuable comments and suggestions. We have now addressed all of the Reviewer’s comments, and provide a point-by-point response below. All changes are underlined in the text of the revised manuscript. Thank you very much.

Comment #2

The title is problematic as it stands. It is particularly long, and does not specify the interest of the study presented. It should be shorter and more concise.

Author’s response:

Thank you for this important point. We have corrected it according to the reviewer’s comment (line 1-4)

Comment #3

The rationale of the study is problematic. Studying the predictive characteristics of lung cancer prognosis is certainly of major interest, but here it is a question of studying the impact of, for example, tumour size on prognosis. However, the impact of such data is already known in the literature, since TNM classifications of tumors are directly linked to patient survival data. Moreover, the 7th classification was used here, whereas the most recent is the 8th; the retrospective nature of the study does not pose any problem in using an up-to-date classification.

Author’s response:

Thank you for this important point. The statistical analysis in this study really focus on the impact of smoking and screened status on lung cancer characteristic and mortality based on four groups. However, from visualization of the graphic abstract (figure 2) shown in discussion section, we have observed this important points about the screening behaviour and socioecological impacts on lung cancer subjects. We have revised the discussion and add the graphic abstract visualization in the discussion section to address this important point and translate the real-world data into clinical practice about strategic policy and educational program to increase smokers’ willingness for lung cancer screening (line 232 to 250).

Comment #4

The statistical analyses are problematic. Indeed, statistical data were obtained from the comparison of data from cohorts of extremely heterogeneous power, with more than 1000 patients in the "no LDCT screening" cohorts, and less than 20 patients in the "LDCT screening" cohorts.

Author’s response:

Thank you for this important point. We have several words to address this limitation in the discussion section to address the study limitation. --- As the number of screen-detected lung cancer’s subjects in smokers is still small, larger studies will be needed to further explore the cause in the future (line 223-225)

Comment #5

The definition of "LDCT screening" is also problematic, as it involves comparing data between patients screened by their own choice, whether or not they are smokers. The screening programs under study do not depend on individual choice, and target mainly smoking patients, which poses a major problem regarding the rationale of the study. Maybe the rationale of the study should be reworked.

Author’s response:

Thank you for this important point. Lung cancer screening criteria was designed for heavy smokers according to the NLST trial. However, the prevalence of non-smoking-related lung cancer is increasing in Asia in recent years according to our previous studies and literatures reviews. Therefore, lung cancer screening program in non-smoking population in Asia targeting at high-risk population is important issue. And lung cancer screening program in Asia is more complex because of enrolled smokers and non-smokers at high risk. In this study, we try to investigate the impact of smoking and screening status on lung cancer screening prognosis and knowledge about the lung cancer screening concepts. The statistical analysis in this study really focus on the impact of smoking and screened status on lung cancer characteristic and mortality based on four groups. However, from visualization of the graphic abstract shown in discussion section, we have observed this important points about the screening behaviour and socioecological impacts on lung cancer subjects. We have revised the discussion and add the graphic abstract visualization in the discussion section to address this important point and translate the real-world data into clinical practice about strategic policy and educational program to increase smokers’ willingness for lung cancer screening (Figure 2, line 232 to 250).

Comment #6

Some formulation are confusing regarding the litterature known datas. For example: "Smoking could cause genetic damage of epithelial cell": could is not appropriated here. This data is well demonstrated in the litterature and seems to be questioned here.

Author’s response:

Thank you for these important points. We have corrected it according to reviewer’s recommendation (line 72).

Comment #7

The conclusions currently presented are confusing, and do not allow a clear message to be drawn from this work, which is nevertheless consequent.

Author’s response:

Thank you for this important point. The statistical analysis in this study really focus on the impact of smoking and screened status on lung cancer characteristic and mortality. However, from visualization of the graphic abstract, we have observed this important points about the screening behaviour and socioecological impacts on lung cancer screening and mortality outcome. We have revised the discussion and add the graphic abstract visualization (figure 2) in the discussion section to address this important point and translate the real-world data into clinical practice about strategic policy and educational program to increase smokers’ willingness for lung cancer screening. In addition, we have added several words to translate real-world knowledge/outcome in this study into personal management/education plan for lung cancer screening according to the personal willingness and socioeconomic status.

(figure 2, line 232 to 250).

Comment #8

Many syntax errors are present.

exemple: "participants who diagnosed with lung cancer in the lung cancer register database" ligne 17 "effectively" instead of "indeed", line 30 "Patients were classified four groups": there is no subject in this sentence

Author’s response:

Thank you for this important point. We have revised the maintext and corrected these errors according to the reviewer’s comments (line 109-110).

Reviewer 2 Report

This study assessed the effect of screening for lung cancer by smoking status and estimated the impact of predictors. I have some points to suggest to improve its readability:

  1. I strongly recommend that the authors use the STROBE consensus for observational studies. Important information to judge the quality of the study is widely missing in this study.

  1. Although I thought that you analyzed the behavior of lung cancer screening when I read the title, you only did estimate the prevalence of lung cancer found by screening and hazard ratio of mortality for predictors of lifestyle and clinical information. You should describe the title appropriately to make the reader easy to understand.

  1. You should include a brief background to ensure the validity of this research and the statistical method used in the analysis. I encourage you not to describe the statement in lines 28-30 because of indirect information derived from this analysis.

  1. In lines 45-47, is the prevalence of non-smoking-related lung cancer high in Taiwan? General information in Taiwan may help to insist on mass LDCT lung cancer screening in this study.  

  1. Materials and Methods: Please add information on the age range of study participants. In line 77, the self-paid LDCT exam is subject to those aged 40-80 years. In non-screening groups, the participants who were not eligible for self-paid screening were included. If you want to discuss willingness to screening, you should have focused on subjects with eligible age for screening.

  1. In lines 77-79, you defined positive smoking history as current smokers or cessation of smoking within the previous 5 years. What was the basis for this categorization? Did you mean that former smokers who have quit more than 5 years were included in the non-smoker's group?

  1. In line 87, you have described the death as clinical information. Did you mean the death from lung cancer or all-cause? If all-cause, I recommend you conduct the analysis focused on death from lung cancer and compare the results with all-cause of death.

  1. If you have missing data on covariates, please describe the number and how to deal.

  1. How did you obtain information on the survival or death of patients other than hospital records? If you have lost to follow-up cases, please describe the number and how to deal.

  1. You should describe the use of variables, whether categorical or continuous. Please add more details if you categorize the variables (e.g., gender (male or female)).

  1. Did you confirm that all variables did not violate the proportional hazard assumption? Age usually violates the proportional hazard assumption with time, especially when using age as continuous.

  1. Please add the information on which statistical software you used in the analysis.

  1. All tables should be presented with a basic description with appropriate superstition so that readers can understand it just by looking at the table.

  1. Please much all decimal points in Tables and text.

  1. The median survival days (time) were different for Tables and text. Please correct.

  1. In lines 126-130, 150-153, and Tables 3 and 5, I could not understand the variable's mean. For example, although you have described that "gender (HR=0.861, P=0.0034) in line127, the reader cannot understand which gender has had lower hazard ratio. It would help if you described what the variable means.

  1. I would like to know the cost of self-paid LDCT. In general, smokers have low socio-economic status. Although educational intervention is important, smokers may not afford to access self-paid screening programs.

  1. You may want to add a general statement that there is a risk in any population-based epidemiological study that residual confounding remains as a limitation.

  1. Was the detection rate of lung cancer by screening in this study compared to in general in Taiwan?

Author Response

TITLE: RE:

RE: jpm-1448189, entitled "Impact of smoking status on lung cancer characteristic and mortality rate between screened and non-screened lung cancer cohorts: willingness and educational concern in lung cancer screening program"

We are grateful to the Reviewers for their valuable comments and suggestions. We have now addressed all of the Reviewer’s comments, and provide a point-by-point response below. All changes are underlined in the text of the revised manuscript.

Reviewer#1

Comment #1

I strongly recommend that the authors use the STROBE consensus for observational studies. Important information to judge the quality of the study is widely missing in this study.

 Author’s response:

We highly appreciate the reviewer’s comment. We have revised the manuscript according to the STROKE checklist and reviewer’s comments. Thank you very much.

Comment #2

Although I thought that you analyzed the behavior of lung cancer screening when I read the title, you only did estimate the prevalence of lung cancer found by screening and hazard ratio of mortality for predictors of lifestyle and clinical information. You should describe the title appropriately to make the reader easy to understand.

Author’s response:

Thank you for this important point. The statistical analysis in this study really focus on the impact of smoking and screened status on lung cancer characteristic and mortality. However, from visualization of the graphic abstract (Figure 2), we have observed this important points about the screening behaviour and socioeconomic impacts on lung cancer screening and mortality outcome. We have revised the discussion and add the graphic abstract visualization in the discussion section to address this important point and translate the real-world data into clinical practice about strategic policy and educational program to improve subjects’ willingness and lung cancer screening coverage in smokers (line 1-4; line 232-248).

Comment #3

You should include a brief background to ensure the validity of this research and the statistical method used in the analysis. I encourage you not to describe the statement in lines 28-30 because of indirect information derived from this analysis.

 Author’s response:

Thank you for this important point. We have added several sentences to address the issue in the instruction section and try to translate the study outcome into real-world clinical practice through direct and indirect information’s addressed in this study (line 51-55)

Comment #4

In lines 45-47, is the prevalence of non-smoking-related lung cancer high in Taiwan? General information in Taiwan may help to insist on mass LDCT lung cancer screening in this study.  

Author’s response:

Thank you for this important point. The prevalence of non-smoking-related lung cancer is increasing in recent years according to our previous studies and literatures reviews. Prevalence of non-smoking-related lung cancer high in Taiwan is around 1.55 to 2.34%. Therefore, lung cancer screening program in non-smoking population in Asia targeting at high-risk population is important issue. We have added several words to address this point in the introduction section (line 66-72).

Comment #5

Materials and Methods: Please add information on the age range of study participants. In line 77, the self-paid LDCT exam is subject to those aged 40-80 years. In non-screening groups, the participants who were not eligible for self-paid screening were included. If you want to discuss willingness to screening, you should have focused on subjects with eligible age for screening

 Author’s response:

Thank you for this important point, the enrolled hospital-based lung cancer cohort was on the age range of 40-80 years. We have added several words to address this point in the study method section (line 89-91).

 Comment #6

In lines 77-79, you defined positive smoking history as current smokers or cessation of smoking within the previous 5 years. What was the basis for this categorization? Did you mean that former smokers who have quit more than 5 years were included in the non-smoker's group?

 Author’s response:

Thank you for this important point, the enrolled hospital-based lung cancer cohort was classified to non-smoker (never smoker) or not according to NHIS definition. We have added several words to address this point in the study method section (line 102-107).

Comment #7

In line 87, you have described the death as clinical information. Did you mean the death from lung cancer or all-cause? If all-cause, I recommend you conduct the  

analysis focused on death from lung cancer and compare the results with all-cause of death.

Author’s response:

Thank you for this important point. In this study, we investigate the lung cancer death as primary outcome/events. We have added several words to address this point in the study method section (line 105-107).

 Comment #8

If you have missing data on covariates, please describe the number and how to deal.

Author’s response:

Thank you for this important point. A total of 210 people have missing data. We removed this group of people and excluded them from the analysis.

Comment #9

How did you obtain information on the survival or death of patients other than

hospital records? If you have lost to follow-up cases, please describe the number and how to deal.

Author’s response:

Thank you for this important point. Information on the survival or death of lung patients are based the hospital-based cancer register data. In our hospital, the cancer register center will regularly follow up and update the latest status of these cancer patients every 6 month interval.

 Comment #10

You should describe the use of variables, whether categorical or continuous.

Please add more details if you categorize the variables (e.g., gender (male or female).

Author’s response:

Thank you for this important point. We have described these variables more clearly in the results section and tables to address this variable impact on the lung cancer mortality (revised table 3, table 5, line 156-162; line 182-188).

Comment #11

Did you confirm that all variables did not violate the proportional hazard

assumption? Age usually violates the proportional hazard assumption with time, especially when using age as continuous.

 Author’s response:

Thank you for this important point. We have made sure that proportional hazard assumption in all variables. The proportional hazards assumption refers to the fact that the hazard functions are multiplicatively related. That is, their ratio is assumed constant over the survival time, thereby not allowing a temporal bias to become influential on the endpoint (line 128-129).

 Comment #12

Please add the information on which statistical software you used in the analysis.

 Author’s response:

Thank you for this important point. We have added several words to address this issue (line 132-133)

 Comment #13

All tables should be presented with a basic description with appropriate superstition so that readers can understand it just by looking at the table.

Author’s response:

 Thank you for this important point. We have added several words in these tables to address the basic description more clearly (revised table 3 and table 5).

Comment #14

Please much all decimal points in Tables and text.

Author’s response:

Thank you for this important point. We have corrected this points according to the reviewer’s recommendation (revised table 1).

  Comment #15

The median survival days (time) were different for Tables and text. Please correct.

Author’s response

Thank you for this important point. We have corrected these mismatches in the tables and text (revised table 1).

 Comment #16

In lines 126-130, 150-153, and Tables 3 and 5, I could not understand the variable's mean. For example, although you have described that "gender (HR=0.861, P=0.0034) in line127, the reader cannot understand which gender has had lower hazard ratio. It would help if you described what the variable means.

Author’s response

Thank you for this important point. We have described these variables more clearly in the results section and tables to address this variable impact on the lung cancer mortality (revised table 3, table 5, line 156-162; line 182-188).

 Comment #17

I would like to know the cost of self-paid LDCT. In general, smokers have low socio-economic status. Although educational intervention is important, smokers may not afford to access self-paid screening programs.

 Author’s response

Thank you for this important point. Through visualization of the graphic abstract in the discussion section, we have observed this important points about the screening behaviour and socioecological impacts on lung cancer screening and mortality outcome. We have revised the discussion and add the graphic abstract visualization (Figure 2) in the discussion section to address this important point and translate the real-world data into clinical practice about strategic policy and educational program to increase smokers’ willingness for lung cancer screening. National health policy is needed to support the lung cancer screening of low-social-economic smokers. Therefore, the high mortality rate in lung cancer subjects with low-social-economic smokers group would be improved ( line 232 to 248).

 Comment #18

You may want to add a general statement that there is a risk in any population-based epidemiological study that residual confounding remains as a limitation.

  Author’s response

Thank you for this important point. We have added several words to address this issue in the discussion section (line 278-280).

Comment #19

Was the detection rate of lung cancer by screening in this study compared to in general in Taiwan?

Author’s response

Thank you for this important point. The prevalence of non-smoking-related lung cancer is increasing in recent years according to our previous studies and literatures reviews. The incidence of lung cancer is similar in the previous studies in Taiwan.

Therefore, lung cancer screening program in non-smoking population in Asia targeting at high-risk population is important issue. We have added several words to address this point in the introduction section (line 66-72).

ments (line 109-110).

Round 2

Reviewer 1 Report

The manuscript has been substantially improved. 
The rationale for the study is now clear.
The results now support the question asked.

Nevertheless, a proofreading by a native English speaker is desirable to improve the syntax.
The formatting and font elements of the characters need to be homogenized and brought in line with the instructions to authors.

Author Response

TITLE: RE:

RE: jpm-1448189, entitled ""

We are grateful to the Reviewers for their valuable comments and suggestions. We have now addressed all of the Reviewer’s comments, and provide a point-by-point response below. All changes are underlined in the text of the revised manuscript.

Reviewer#1

Comment #1 The manuscript has been substantially improved. The rationale for the study is now clear.The results now support the question asked. Nevertheless, a proofreading by a native English speaker is desirable to improve the syntax. The formatting and font elements of the characters need to be homogenized and brought in line with the instructions to authors.

Author’s response:

We highly appreciate the reviewer’s comment. We choose fast MDPI's English Editing service, and revised the manuscript according to recommendation of MDPI’s English editing and the native English author (Shu-Ching Yang). Thank you very much.

Reviewer2

Comment #1

The authors responded appropriately to my comments and their manuscript improved tremendously. Please make sure that the median and range of age at diagnosis is correct, add a unit and clarify what the value in parentheses in Table 1.  Thank you for giving me the opportunity to review your paper.

Author’s response:

We highly appreciate the reviewer’s comment. We have revised table 1 according to reviewer’s comment. In addition, we make final English editing according to recommendation of MDPI’s English editing and the native English author (Shu-Ching Yang). Thank you very much.

Reviewer 2 Report

The authors responded appropriately to my comments and their manuscript improved tremendously. Please make sure that the median and range of age at diagnosis is correct, add a unit  and clarify what the value in parentheses in Table 1.  Thank you for giving me the opportunity to review your paper.

Author Response

(The authors gave the same response as above.)
